# Reliability and Validity of the Korean Version of the High-Performance Work System Scale (HPWS-K)

**DOI:** 10.3390/ijerph192013708

**Published:** 2022-10-21

**Authors:** Hyesun Kim, Kawoun Seo, Taejeong Jang

**Affiliations:** 1Department of Nursing, Hyejeon College, Hongseong 32244, Korea; 2Department of Science of Nursing, Joongbu University, Geumsan 32713, Korea; 3Department of Nursing, Woosuk University, Wanju-gun 55338, Korea

**Keywords:** high-performance work systems (HPWS), patient safety, health care environment, nursing management

## Abstract

The application of the concept of high-performance work system (HPWS) to nurses can improve the quality of nursing care by enhancing nurse supply and patient safety culture. This study aimed to test the validity and reliability of the Korean version of the HPWS Scale (HPWS-K) and verify its adequacy for measuring HPWS within the context of nurses working in Korean hospitals. Data analyses were performed by using data from 214 nurses engaged in patient care in Korean general hospitals with over 300 beds, with SPSS version 22.0 and AMOS version 26.0 software. The data collected were translated into Korean and the adapted Korean version was back-translated into English by bilingual nursing professionals. The content validity of the HPWS-K was tested by administering it to 10 participants, and the collected data underwent reliability and validity testing. In the exploratory factor analysis, the total variance explained was 49.97%, and the instrument’s reliability (Cronbach’s α) was 0.87. A one-factor confirmatory factor analysis showed an acceptable model fit. The results confirmed the feasibility of using the HPWS-K with Korean nurses, and its application in this context is expected to contribute to creating a safe and effective healthcare environment in Korea.

## 1. Introduction

The high-performance work system (HPWS) is a concept in human resources (HR) management; it was designed to enhance organizational performance through organizational members’ autonomous job crafting by continuously inducing them to develop their job competence and motivation [1]. Indeed, the literature shows that the components of HPWS, namely recruitment, education and training, personnel rating, compensation, and job security [2], positively affect organizational performance by improving organizational members’ abilities, attitudes, and motivation [3,4]. Researchers have also described how HPWS is being applied to various industries, providing immediate responses to consumer demands, and strengthening competitiveness in the current, rapidly changing corporate environment of the information age [5].

Hospitals have also been applying fast-paced changes and coping strategies to tackle the novel challenges evoked by the new consumer-centered healthcare market, with HPWS and its proven effectivity having become a standard strategy for improving the design and operation of healthcare providers [6]. Meanwhile, research shows that nurses make up the largest proportion of the healthcare workforce and that hospital management stakeholders are experiencing difficulties adequately supplying nurses in relation to their frequent turnover [7]. Considering that nurse workforce shortages and the frequent replacement of nurses greatly affects the quality of healthcare services experienced by patients [8], finding better solutions, such as creating an optimal work system in hospitals, instead of relying on hiring new nurses to replace the nurses who leave [9], may be more effective and even necessary.

Researchers have applied HPWS to healthcare settings, finding that an effective HPWS improved the organizational commitment of hospital employees [10], lowered turnover intention [11], and improved patient safety by reducing medical error frequency [12]. Thus, if HPWS is applied to nursing settings, it may reduce nurse turnover [13], enhance their organizational commitment [14], contribute to patient safety [15], and improve the quality of nursing performance provided to patients. These enhancements may lead to improvements in hospital performance (Figure 1).

In South Korea, to date and to the best of our knowledge, the bulk of research on HPWS has been related to non-healthcare company employees, production workers, and airline cabin crews [16,17,18], whereas a small number of researchers have studied this concept with medical doctors in hospital settings [19]. In particular, there is currently a lack of reliable, validated tools for assessing HPWS in nursing-related healthcare settings. In this regard, previous studies have used the self-reported 10-item HPWS Scale to measure HPWS. This scale was developed by Etchegaray and Tomas [20] in the United States of America, based on a literature review and hospital executive ratings. Significantly, in the study of hospital executives, HPWS was found to be a reliable and predictable variable in relation to patient safety. It assesses HPWS-related aspects including educational opportunities, rewards, information provision, teamwork, employee surveys, and job security, and its validity and reliability were established at the time of development [20]. Moreover, high-performance work systems allow employees to focus more on their work and maintain a high level of work performance [21]. Therefore, in this study, the simultaneous validity was measured by using the high-performance work system, and the work performance evaluation scale was used as a tool to measure the simultaneous effectiveness.

### Purpose

This study aimed to test the validity and reliability of the Korean version of the HPWS Scale (HPWS-K) and evaluate its adequacy as an instrument to measure HPWS within the context of Korean nurses working in clinical settings. To achieve this goal, three objectives were set:
translating the HPWS Scale into Korean;testing the validity (face validity, construct validity, and concurrent validity) of the HPWS-K; andtesting the reliability of the HPWS-K.

## 2. Materials and Methods

### 2.1. Study Design

This study was a methodological, instrument development study.

### 2.2. Participants

The inclusion criteria for nurses in this study were as follows: (1) currently working in a Korean general hospital with over 300 beds; (2) having at least three months of work experience in the current hospital; (3) understanding the study purposes and voluntarily agreeing to participate. The exclusion criteria were nurses in units that do not work directly with patients, such as central supply rooms.

In total, 220 questionnaire forms were distributed based on the minimum sample size (*n* = 200) required for confirmatory factor analysis for construct validity testing [22] and a 10% expected retrieval rate. After removing the data from six questionnaires with incomplete answers, data from 214 questionnaires were used for EFA and CFA analyses.

### 2.3. Survey Instruments

#### 2.3.1. High Performance Work Systems Scale

Prior to the validity and reliability testing of the HPWS-K, permission to translate the original scale and use the translated version was obtained from the authors of the original scale, namely Etchegaray and Thomas [20]. The 10-item HPWS Scale is rated on a 5-point Likert scale ranging from 1 (strongly disagree) to 5 (strongly agree), This tool consists of 10 questions about training, compensation, information sharing, teamwork, communication, technical education, job security, decision-making, and recycling. It ranges from a minimum of 5 to a maximum of 50 points. Cronbach’s α was 0.92 at the time of development, and 0.87 in this study.

#### 2.3.2. Task Performance Evaluation Instrument

Task performance was measured using the Task Performance Evaluation Instrument (TPEI) developed by Paik et al. [23]. This tool was used by the manager to evaluate the job performance of clinical nurses in the clinical field. According to previous studies, a high-level of work performance is highly correlated with high-performance work systems [21], so the validity of high-performance work system tools can be evaluated through simultaneous work performance evaluation. This 35-item scale comprises four task-related factors: knowledge (8 items), attitude (13 items), performance (7 items), and ethics (7 items). Each item is rated on a 5-point Likert scale ranging from 1 (very unlikely) to 5 (very likely), with higher total scores indicating higher task performance levels. The tool ranges from a minimum of 35 to a maximum of 175 points. Cronbach’s α values ranged from 0.918 to 0.954 at the time of development, and it ranged from 0.927 to 0.968 in this study.

#### 2.3.3. Sociodemographic Variables

The general characteristics of age, gender, education level, marital status, religious status, current position, and nurse career were investigated. Satisfaction with nurses and satisfaction with the working environment were investigated by using a 3-point Likert scale ranging from 1 (Unsatisfied) to 3 (Satisfied).

### 2.4. Research Process

#### 2.4.1. Translation/Back-Translation Phase

Translation of the questionnaire items: three bilingual nursing professionals (two nursing professors and one clinical nurse) separately performed a double translation of the items of the HPWS Scale into Korean, compared the translations among themselves, and modified and supplemented the Korean version.Back-translation and confirmation of the translated items: a bilingual nursing professor back-translated the Korean translation of the HPWS Scale items into English, which was then checked against the original questionnaire by a second nursing professor to ensure that the meaning of each item was accurately captured. The final items were established after the accuracy of the final translated version was confirmed in a discussion with all those involved.

#### 2.4.2. Content Validity Testing

To rate the understandability of each item of the HPWS-K, the content validity index (CVI) of the factors (e.g., comprehensiveness and time spent on completing the scale) was computed through the participation of 10 nurses with a Master’s or doctoral degree in nursing and more than 10 years of clinical experience. In the content validity verification conducted during the whole month of July 2021, difficult words or incomprehensible phrases were revised without changing the overall meaning. Each item was rated on a 4-point Likert scale ranging from 1 (no relevance) to 4 (high relevance).

#### 2.4.3. Construct Validity Testing

The construct validity of the HPWS-K was tested by performing item analysis, exploratory factor analysis (EFA), and confirmatory factor analysis (CFA). Item analysis was performed to calculate the correlation coefficients between all revised items, indicating the relationships between individual items and the scale; following the criteria used in previous research [24], values greater than or equal to 0.3 were selected. In the EFA, which was conducted for factor extraction, a factor analysis method without factor rotation was used because the scale comprised a single-factor structure in prior research. Factors with the total variance explained (TVE) exceeding 50% [25] and eigenvalues of 1.0 or greater were extracted, which were then checked for factor loadings in excess of 0.4.

#### 2.4.4. Concurrent Validity Testing

Drawing on the finding of a previous study, which showed that the HPWS Scale is highly correlated with the TPEI in nurses [26], concurrent validity testing was conducted by calculating the correlation coefficients between the HPWS-K and TPEI.

#### 2.4.5. Reliability Analysis

The reliability of the HPWS-K was tested by calculating Cronbach’s α, which is an index of reliability for internal consistency.

### 2.5. Data Collection and Ethical Considerations

Prior to proceeding with the study, approval was obtained for the study protocol from the Institutional Review Board of Joongbu University (approval number: JIRB-2021050302-03). Permission for data collection was obtained by the researcher, who visited the nursing departments of three hospitals in D region. The study’s purpose was explained to ward nurses, and the study was conducted after obtaining written consent from those who agreed to participate in the study. Data were collected in August 2021. Data analysis was conducted from September to December 2021. After being given explanations about the study’s purpose, written consent was obtained from 220 ward nurses. The informed consent form contained explanations about the survey’s purpose, anonymity, confidentiality, exclusive use of the data for study purposes, and storage of the filled-in questionnaires on a computer under lock and key for three years, followed by disposal. The link to the survey generated in Google Forms was distributed to the participants who completed the questionnaire online. Those who completed the questionnaire were given a beverage coupon of appreciation.

### 2.6. Data Analysis

The collected data were analyzed by using SPSS for Windows version 22.0 and AMOS version 26.0 software. To analyze the participants’ general characteristics, basic descriptive statistics (i.e., mean and standard deviation) were calculated. For content validity testing, the item content validity index (I-CVI) was computed by using a 4-point Likert scale with the help of an expert group. EFA and CFA were performed to assess construct validity. To evaluate EFA’s model fit, Kaiser–Meyer–Olkin (KMO) value, Bartlett’s test of sphericity, and principal component analysis with varimax rotation were performed [22].

Based on the analysis results, factors were extracted according to the criteria of eigenvalue ≥1, communality ≥0.40, and the standard factor loading of 0.50 [25]. The CFA was performed based on the results of EFA, and the model fit was tested by using *χ*^2^/df, goodness of fit index (GFI), root mean square residual (RMR), root mean square (RMSEA), comparative fit index (CFI), Tucker–Lewis index (TLI), and incremental fit index (IFI). For concurrent validity testing, Pearson’s correlation coefficients were calculated between the scores for the HPWS-K and TPEI, and instrument reliability was tested by assessing internal consistency with the Cronbach’s α value.

## 3. Results

### 3.1. Participants’ General Characteristics

The participants’ mean age was 33.63 ± 6.08, and their sex distribution was 2% male (*n =* 5) and 98% female (*n =* 209). A Bachelor’s degree was the most common education level (*n =* 156, 72.5%), followed by an Associate’s degree (*n =* 33, 15.7%), Master’s degree (*n =* 23, 10.8%), and doctoral degree or higher (*n =* 2, 1%). Furthermore, 114 nurses (52.9%) were married, and 136 (63.7%) had a religion. The majority of them (84.3%) were general nurses, and the most common work experience was 5 to 10 years (*n =* 70, 32.4%). Additionally, 142 nurses (66.7%) were moderately satisfied with being a nurse. Finally, 117 (54.9%) and 61 (28.4%) nurses found the working environment moderately satisfactory and unsatisfactory, respectively (Table 1).

### 3.2. Validity Analysis

#### 3.2.1. Content Validity

All the 10 items in the HPWS-K showed an I-CVI that exceeded 0.7. Based on the expert opinions collected, Item 3, “Hospital nurses are given information necessary for providing good nursing care,” was revised to “Hospitals provide necessary information for nurses to provide good nursing care”. Furthermore, item 6 was revised from “Nurses at my hospital receive performance appraisals that help them improve their performance” to “Nurses at my hospital receive performance appraisals, which help them improve their nursing performance”.

#### 3.2.2. Construct Validity: Item Analysis

Each item of the HPWS-K was analyzed for construct validity, and the corrected item-total correlation coefficients ranged from 0.30 to 76. Because no item scored less than 0.3, which is the cutoff point [24], the EFA was performed with all 10 items. Skewness and kurtosis were used to assess data normality. The skewness and kurtosis for the items ranged from −0.69 to 0.37 and −0.31 to 0.89, respectively. Hence, they were above the absolute values of 2 and 7, demonstrating data normality. The mean values by item ranged from 2.51 to 3.57, and the standard deviation ranged from 0.81 to 1.05., with the mean and standard deviation for the total scores standing at 3.20 and 0.91, respectively (Table 2).

#### 3.2.3. Construct Validity: Exploratory Factor Analysis

The KMO value assessed prior to EFA stood at 0.87, and the Bartlett’s test of sphericity at *χ*^2^ = 426.93 (*p* < 0.001), demonstrating EFA’s adequacy [24]. The EFA conducted by performing principal component analysis confirmed the initial eigenvalue of 1.0 or greater, and the factor loadings of items ranged from 0.53 to 0.84, with the eigenvalue and explanatory power calculated at 4.76 and 49.97%, respectively (Table 2).

#### 3.2.4. Construct Validity: Confirmatory Factor Analysis

The CFA was performed by using the 10 items identified in the EFA. The model fit was found to be acceptable, with *χ*^2^/df = 2.09, RMR = 0.06, CFI = 0.90, and IFI = 0.91 (Table 3; Figure 2). The single-factor instrument was named HPWS-K (the Korean version of the High-Performance Work Systems Scale).

#### 3.2.5. Concurrent Validity

The correlation between HPWS-K and TPEI was assessed to test the criterion validity of the HPWS-K. As a result, the HPWS-K was found to be significantly positively correlated with the TPEI (r = 0.53, *p* < 0.001). HPWS-K was also found to be significantly positively correlated with each of the four TPEI factors as follows: knowledge (r = 0.49, *p* < 0.001), attitude (r = 0.51, *p* < 0.001), performance (r = 0.49, *p* < 0.001), and ethics (r = 0.52, *p* < 0.001; Table 4).

#### 3.2.6. Reliability Analysis

The overall Cronbach’s α of the HPWS-K was 0.88 (Table 2). The reliability of an instrument is considered excellent at the Cronbach’s α value of 0.90 or higher, good at 0.80 or higher, acceptable at 0.70 or higher, and unreliable at 0.50 or lower [22].

## 4. Discussion

This study aimed to verify a high-performance work system tool and apply it to the medical environment in Korea by evaluating its validity in terms of improving patient safety and work performance ability in a medical environment. Based on analyses, it established that high-performance work system tools are useful for predicting task performance.

This study was conducted to test the reliability and validity of the HPWS-K, which is the translated version of the HPWS Scale developed by Etchegaray and Thomas [20], to objectively measure HPWS as perceived by nurses working in general hospitals and within the context of patient safety. The Cronbach’s α value of the HPWS-K was 0.88, which is lower than that at the time of development (0.92) but similar to that of a previous study on nurses [27] and higher than that of the German version of the HPWS Scale (0.85) [28]. Furthermore, the Cronbach’s α of the HPWS-K exceeded 0.70, which was proposed as the cutoff for acceptable internal consistency of a psychometric instrument in prior research [24]. Accordingly, although the HPWS-K showed good internal consistency, its reliability must be reaffirmed by applying it to Korean nurses in replication research.

The content validity evaluation of the HPWS-K was performed to assess the applicability of this instrument, which was originally developed in a foreign country, to Korean hospital nurses. The mean I-CVI of the 10-item HPWS-K was 0.72, demonstrating its content validity. The HPWS was developed for hospital management in the original study [20] but has been applied to nurses in previous research, and its content validity in the current study demonstrated its applicability to Korean nurses.

To test the construct validity of the HPWS-K, EFA, and CFA were performed. The EFA results showed a single-factor structure for the 10 items, and the overall explanatory power was calculated at 49.97%, demonstrating the instrument’s construct validity. Additionally, the factor load of teamwork in this study was 0.53, which was lower than those of other factors, i.e., 0.61 to 0.84. This is similar to the result of removing teamwork items for the same reason in previous studies. However, teamwork is an important indicator in the medical environment, especially in the evaluation of nursing work performance [7], and in this study, the factor load was 0.53, which is higher than the reference value [22]. Therefore, 10 questions are considered appropriate as long as the teamwork item is included. Because CFA is required when an original instrument with a proven validity is translated into a foreign language [29], CFA was also performed to test the construct validity by assessing the model fit. Model fit testing of the 10 items in a one-factor model resulted in the following values: *χ*^2^/df = 2.09, RMR = 0.06, CFI = 0.90, and IFI = 0.91. These values demonstrated the suitability of the 10-item HPWS-K for application to Korean nurses. These results are supported by those in the original development study of the HPWS Scale and the study that developed the German version of the HPWS Scale [20,28], with both showing a single-factor structure for the scale.

The correlation coefficients between the HPWS-K and TPEI, which demonstrate concurrent validity, ranged from 0.49 to 0.53, demonstrating statistically significant correlations between the HPWS-K and all four factors of the TPEI (knowledge, attitude, performance, and ethics). These results are similar to those of a previous study [12], which demonstrated that the efficient application of HPWS positively affects patient safety by enhancing nursing performance. In addition, the positive correlation between the HPWS-K and the attitude subscale of the TPEI concurs with the evidence in one prior study [11], which showed that a well-established HPWS reduces turnover intention in nurses. The HPWS-K was thus judged to be suitable for measuring nurses’ job performance. Accordingly, this study’s results may serve as a basis for designing strategies related to nursing workforce management by assessing HPWS and its correlation with nursing task performance. In summary, as suggested in the previous literature review, the high-performance work system increases employees’ focus on work performance and leads to high work performance. In the context of nurses, it contributes to high nursing work performance and establishes a patient safety culture in a medical environment. This study’s results are in line with those of preceding studies. Thus, the tool tested will be useful for improving the quality of nursing services and establishing a patient safety culture through systematic nursing work in a rapidly changing medical environment.

In this study, the reliability and validity of the HPWS-K were tested to determine its applicability to nurses. The results showed that it is reliable, valid, and comprises 10 short, easy-to-answer, and easy-to-access items appropriate for a self-reported questionnaire. The HPWS-K can also be compared with other translated versions of the HPWS Scale because it was translated directly from the original instrument. This study’s results can thus be used as basic data for creating environments propitious to patient safety and are expected to provide basic data for replication research on nurses to confirm the reliability and validity of the instrument, allowing it to be applied to nursing research and nursing policy in future.

## 5. Conclusions

This study tested the reliability and validity of the HPWS-K by applying it to nurses working in hospitals. Based on analyses, its validity (content, construct, and concurrent validity) and reliability were established, confirming its appropriateness for application to Korean nurses. This study is expected to contribute to nursing policy and patient safety culture by allowing for stakeholders to measure HPWS within the working contexts of Korean nurses.

The study’s findings allow for the following recommendations to be made. First, the results of applying the HPWS-K in this study can be used as basic data for developing patient safety education programs. Secondly, the correlation between HPWS and nursing job performance can be performed, and the related results be used for setting up strategies for nursing workforce management. Thirdly, any interpretations and generalizations of this study’s results to nurses working in other regions of South Korea should be made with caution, and this suggests that replication research could be conducted to allow for the findings to be generalized to other settings.

## Figures and Tables

**Figure 1 ijerph-19-13708-f001:**
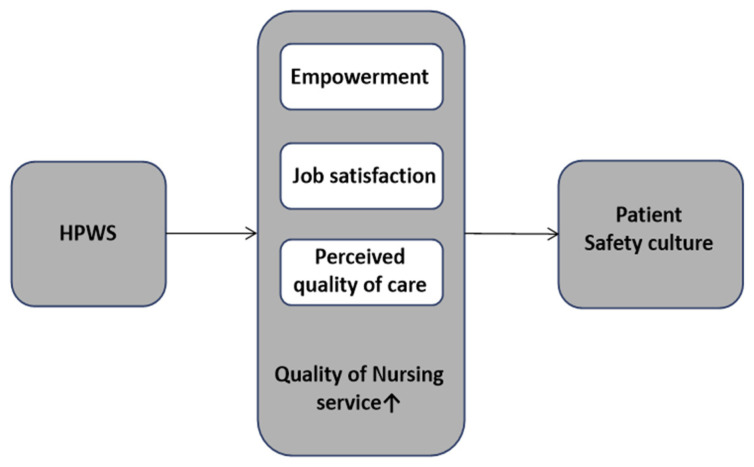
Conceptual framework from the literature review.

**Figure 2 ijerph-19-13708-f002:**
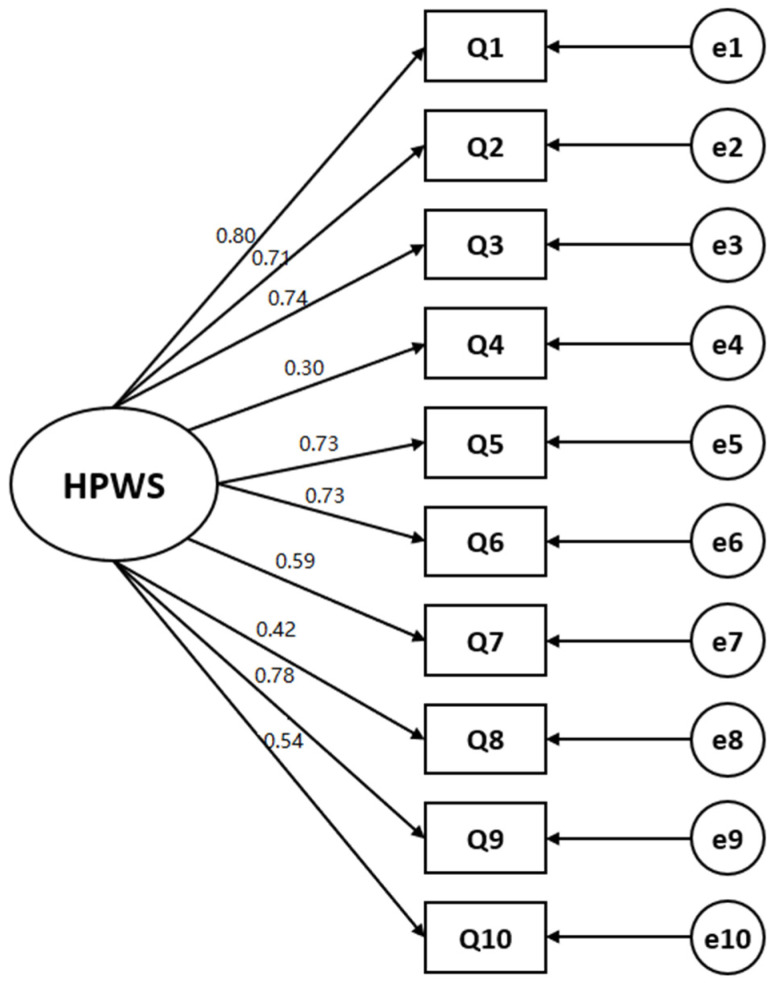
Measurement model of the Korean version of the HPWS Scale (HPWS-K).

**Table 1 ijerph-19-13708-t001:** General characteristics of participants (*N =* 214).

Characteristics	Categories	*N* (%) or Mean ± Standard Deviation
Age (years)		33.63 ± 6.08
Sex	Male	5 (2)
	Female	209 (98)
Education level	Degree	33 (15.7)
	Bachelor’s degree	156 (72.5)
	Master’s degree	23 (10.8)
	Doctoral degree	2 (1.0)
Marital status	Single	100 (47.1)
	Married	114 (52.9)
Having a religion	Yes	136 (63.7)
	No	78 (36.3)
Current position	General nurse	180 (84.3)
	Charge nurse	29 (13.7)
	Higher than head nurse	5 (2.0)
Work experience as a nurse	5 years or less	42 (19.6)
	5–10 years	70 (32.4)
	10–15 years	46 (21.6)
	Over 15 years	56 (26.5)
Satisfaction as a nurse	Unsatisfied	36 (16.7)
	Moderate	142 (66.7)
	Satisfied	36 (16.7)
Satisfaction with the working environment	Unsatisfied	61 (28.4)
	Moderate	117 (54.9)
	Satisfied	36 (16.7)

**Table 2 ijerph-19-13708-t002:** Item analysis and exploratory factor analysis of the Korean version of the HPWS Scale (HPWS-K; *N =* 214).

Items	Mean ± Standard Deviation	Skewness	Kurtosis	Corrected Item Total Correlation	Communalities	Factor Loading
1. Employees in my hospital area are provided opportunities to learn new skills. (Skills)	3.13 ± 0.82	−0.02	0.10	0.72	0.69	0.78
2. Employees in my hospital area are given rewards for doing a good job. (Rewards)	2.51 ± 1.05	0.37	−0.16	0.59	0.76	0.74
3. Employees in my hospital area receive the necessary information to do a good job. (Information)	3.03 ± 0.84	0.05	0.20	0.67	0.62	0.78
4. Teamwork is important for providing quality service to patients. (Teamwork)	4.00 ± 0.81	−0.69	0.89	0.42	0.66	0.53
5. Employees in my hospital area are asked how workplace processes can be improved. (Workplace)	3.28 ± 0.88	−0.51	0.29	0.69	0.60	0.76
6. Employees in my hospital area receive performance appraisals that help them improve their performance. (Appraisal)	3.06 ± 0.88	−0.30	0.06	0.65	0.57	0.77
7. Employees in my hospital area receive training on quality improvement methods. (Quality)	3.50 ± 0.83	−0.42	0.56	0.57	0.48	0.70
8. Employees in my hospital area have job security. (Job Security)	3.57 ± 1.02	−0.36	−0.31	0.50	0.54	0.61
9. Employees in my hospital area see improvements in this hospital area based on the results of employee surveys. (Survey)	2.98 ± 0.98	−0.02	0.01	0.76	0.68	0.84
10. The best candidate for the job is hired in this hospital area. (Candidate)	2.97 ± 0.95	−0.08	−0.06	0.52	0.37	0.61
Mean ± Standard deviation total	3.20 ± 0.91	−0.20	0.16			
Eigenvalue						4.76
Variance (%)						49.97
Cumulative variance (%)						49.97
Cronbach’s α						0.882
Kaiser-Meyer-Olkin (KMO)						0.881
Bartlett’s test of sphericity					*χ*^2^ = 769.83 (*p* < 0.001)	

HPWS, high-performance work systems.

**Table 3 ijerph-19-13708-t003:** Fit indices for the confirmatory factor analysis (*N =* 214).

Variables	CMIN/df	GFI	RMR	RMSEA	CFI	TLI	IFI
Evaluation criteria	≤3	≥0.90	≤0.05–0.08	≤0.05–0.08	≥0.90	≥0.90	≥0.90
HPWS	2.09	0.88	0.06	0.10	0.90	0.88	0.91

**Table 4 ijerph-19-13708-t004:** Correlation among concurrent variables (*N* = 214).

Variables	High Performance Work System Scale Correlation (r)										
r (*p*)	Total	Item1	Item2	Item3	Item4	Item5	Item6	Item7	Item8	Item9	Item10
TPEI (total)	0.53 **	0.42 **	0.25 **	0.39 **	0.42 **	0.37 **	0.34 **	0.43 **	0.41 **	0.39 **	0.32 **
TPEI-Knowledge	0.49 **	0.38 **	0.28 **	0.33 **	0.39 **	0.26 **	0.25 **	0.37 **	0.43 **	0.27 **	0.29 **
TPEI-Attitude	0.51 **	0.38 **	0.28 **	0.37 **	0.36 **	0.41 **	0.33 **	0.38 **	0.32 **	0.38 **	0.34 **
TPEI-Performance	0.49 **	0.42 **	0.22 **	0.36 **	0.39 **	0.30 **	0.30 **	0.40 **	0.39 **	0.39 **	0.30 **
TPEI-Ethics	0.52 **	0.32 **	0.20 **	0.31 **	0.37 **	0.33 **	0.34 **	0.40 **	0.36 **	0.39 **	0.31 **

** *p*< 0.001, TPEI = Task Performance Evaluation Instrument.

## Data Availability

Not applicable.

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
