# Peer review of "Reliability and Validity of the Korean Version of the High-Performance Work System Scale (HPWS-K)"

_ijerph, 2022, doi:10.3390/ijerph192013708_

Round 1

Reviewer 1 Report

This article is about validating the validity and reliability of the Korean version of the High-Performance Work System scale and testing its adequacy for measuring high-performance work systems in the context of nurses working in Korean hospitals. The relevance of the study is justified by the fact that applying the concept of a high-performance work system to nurses can improve the quality of nursing care by increasing the number of nurses and the culture of patient safety. Data analysis was performed using data from 214 nurses caring for patients in Korean general hospitals with over 300 beds using SPSS version 22.0 and AMOS version 26.0 software. The collected data was translated into Korean, and the adapted Korean version was again translated into English by bilingual nurses. The validity of the content of the high-performance system of work was verified by introducing it to 10 participants, and the collected data was tested for reliability and validity. In exploratory factor analysis, the total explained variance was 49.97% and the reliability of the instrument (Cronbach's α) was 0.87. One-way confirmatory factor analysis showed an unacceptable fit to the model.

Despite the satisfactory quality of the article, some shortcomings need to be corrected.

  1. The questionnaire should be described in more detail and grounded.
  2. It is recommended to visualize the results of the survey. It will increase the quality of the paper.
  3. The Discussion section should be slightly expanded because it is the main finding of the paper.
  4. The outcomes of the research should be defined.
  5. The practical and scientific novelty of the paper should be highlighted.

In summarizing my comments I recommend that the manuscript is accepted after major revision, including the Discussion section expanding.

Author Response

Dear Reviewer,
Thank you for giving me the opportunity to submit a revised draft of my manuscript titled “Reliability and Validity of the Korean Version of the High-Performance Work System Scale (HPWS-K)” to International Journal of Environmental Research and Public Health. We appreciate the time and effort that you and the reviewers have dedicated to providing your valuable feedback on my manuscript. We are grateful to the reviewers for their insightful comments on my paper. We have been able to incorporate changes to reflect most of the suggestions provided by the reviewers. We have highlighted the changes within the manuscript.
Here is a point-by-point response to the reviewers’ comments and concerns.

1.The questionnaire should be described in more detail and grounded.
[Re:] Thank you for your suggestion. We have revised the questionnaire based on your suggestion. 
(page 3, lines 103-106, 109-113, 116-117, 122-123)
2. It is recommended to visualize the results of the survey. It will increase the quality of the paper.
[Re:] Thank you for this insightful suggestion. We have added this based on your suggestion. 
(page 7-8 lines 246-247)
3. The Discussion section should be slightly expanded because it is the main finding of the paper. The outcomes of the research should be defined. The practical and scientific novelty of the paper should be highlighted. In summarizing my comments I recommend that the manuscript is accepted after major revision, including the Discussion section expanding.
[Re:] Thank you for your careful and thorough review of our paper, for which we are truly grateful. Thank you for the suggestion regarding the Discussion section. Following your suggestion, we have revised it extensively. 
(page8-9, lines 252-255, 274-280, 299-304)

Reviewer 2 Report

Thank you for the opportunity to review this study. These are my comments:

2.2 The exclusion criteria in the Materials and Methods are not exclusion criteria. This is just the opposite of the inclusion criteria. Revise it.

2.3.3 How did you measure satisfaction?

Author Response

Dear Reviewer,
Thank you for giving me the opportunity to submit a revised draft of my manuscript titled “Reliability and Validity of the Korean Version of the High-Performance Work System Scale (HPWS-K)” to International Journal of Environmental Research and Public Health. We appreciate the time and effort that you and the reviewers have dedicated to providing your  valuable feedback on my manuscript. We are grateful to the reviewers for their insightful 
comments on my paper. We have been able to incorporate changes to reflect most of the suggestions provided by the reviewers. We have highlighted the changes within the manuscript.
Here is a point-by-point response to the reviewers’ comments and concerns.

point 1. The exclusion criteria in the Materials and Methods are not exclusion criteria. This is just the opposite of the inclusion criteria. 
[Re:] Thank you for your this comment. We have revised the exclusion criteria accordingly. 
(page 3, lines 88-89 )

2. How did you measure satisfaction?
[Re:] Thank you for this query. We have explained this in the revised manuscript. 
(page 3, lines 112-113 )

Reviewer 3 Report

The paper is very informative and provides a valuable source document for anyone who needs to know and understand this issue. However, some changes are needed:

·         Line 61-64: The sentence is not clear. Please, rewrite it.

·         Line 85-86: Give rationale with references for "300 beds" and "3 months".

·         Line 170-171: Use the proper citations when criterion or cut-off are defined.

·         Line 182-183: Discuss the gender distribution and correct the numbers in parentheses to add up to 214.

·         Line 234: Are you performing the Confirmatory Factorial analysis on the same sample? Please define.

Author Response

Dear Reviewer,
Thank you for giving me the opportunity to submit a revised draft of my manuscript titled “Reliability and Validity of the Korean Version of the High-Performance Work System Scale (HPWS-K)” to International Journal of Environmental Research and Public Health. We appreciate the time and effort that you and the reviewers have dedicated to providing your valuable feedback on my manuscript. We are grateful to the reviewers for their insightful comments on my paper. We have been able to incorporate changes to reflect most of the suggestions provided by the reviewers. We have highlighted the changes within the manuscript.
Here is a point-by-point response to the reviewers’ comments and concerns.

point 1. Line 61-64: The sentence is not clear. Please, rewrite it.
[Re:] Thank you for pointing out this issue, for which we are apologetic. We have revised the sentence to enhance clarity.
(page 2, lines 61-66)

point 2. Line 85-86: Give rationale with references for "300 beds" and "3 months".
[Re:] Thank you for this insightful suggestion. We have deleted the information and revised it to include more appropriate information. 
(page 3, lines 88-89)

point 3. Line 170-171: Use the proper citations when criterion or cut-off are defined.
[Re:] Thank you for your suggestion. We have revised the relevant citations accordingly. 
(page 4, lines 171)

point 4. Line 182-183: Discuss the gender distribution and correct the numbers in parentheses to add up to 214.
[Re:] Thank you for the suggestions. We have discussed the gender distribution and corrected the numbers in parentheses. 
(page 5, lines 182-190)

point 5. Line 234: Are you performing the Confirmatory Factorial analysis on the same sample? Please define.
[Re:] Thank you for your question and suggestion. We have defined it in the “Participants”section in Materials and Methods.
(page 3, lines 93)

Reviewer 4 Report

Dear authors,

Thank you for the opportunity to read the exciting article “Reliability and Validity of the Korean Version of the High-Performance Work System Scale (HPWS-K)”. In my opinion, the paper aligns well with the aims and scope of the journal as it addresses an important and highly relevant topic in the field of health sciences, quality of care and patient safety. The article is described in a scientifically sound manner and leaves only a few questions unanswered.

Some methodological issues could have been addressed in more detail to avoid incorrect results and conclusions in the study. The appropriate items are listed below.

L 103: In the methods, the TPEI is described as a tool to measure concurrent validity. This comes as a complete surprise in this section. I would recommend mentioning the TPEI already in the introduction and referring to comparable studies.

L 127: When did the content validation take place? How were the nurses recruited?

L 170: Maybe you can correct Kaiser-Meyer-Olkin (you have been writing Keiser)?

L 152: The study fails to describe how access to hospitals and recruitment of nurses was done. A more concrete description would be helpful in this section.

L 182: In my opinion, the data in the text do not match the data in the table (e.g., 2% male, n= 5; 98% female, n= 209, bachelor/master). I would recommend a re-check of the data.

Author Response

Dear Reviewer,
Thank you for giving me the opportunity to submit a revised draft of my manuscript titled “Reliability and Validity of the Korean Version of the High-Performance Work System Scale (HPWS-K)” to International Journal of Environmental Research and Public Health. We appreciate the time and effort that you and the reviewers have dedicated to providing your valuable feedback on my manuscript. We are grateful to the reviewers for their insightful comments on my paper. We have been able to incorporate changes to reflect most of the suggestions provided by the reviewers. We have highlighted the changes within the manuscript.
Here is a point-by-point response to the reviewers’ comments and concerns.

point 1. L 103: In the methods, the TPEI is described as a tool to measure concurrent validity. This comes as a complete surprise in this section. I would recommend mentioning the TPEI already in the introduction and referring to comparable studies.
[Re:] Thank you for your valuable recommendation, based on which we have revised the information regarding TPEI. 
(page 2, lines 68-72)

point 2. L 127: When did the content validation take place? How were the nurses recruited?
[Re:] Thank you for your queries. We have explained these aspects in the revised manuscript. 
(page 4, lines 159)

point 3. L 170: Maybe you can correct Kaiser-Meyer-Olkin (you have been writing Keiser)?
[Re:] Thank you for your pointing this out. We have revised the spelling based on your comment. 
(page 4, lines 171)

point 4. L 152: The study fails to describe how access to hospitals and recruitment of nurses was done. A more concrete description would be helpful in this section.
[Re:] Thank you for this insightful suggestion. We have revised the relevant description accordingly. 
(page 4, lines 156-158)

point 5. L 182: In my opinion, the data in the text do not match the data in the table (e.g., 2% male, n= 5; 98% female, n= 209, bachelor/master). I would recommend a re-check of the data.
[Re:] Thank you for your pointing out the issue regarding data mismatch, for which we apologize. We have rechecked the data and revised them as necessary. 
(page 5, lines 182-190)

Round 2

Reviewer 1 Report

Thanks for the authors for their analysis and considering the recommendations of the reviewers. However, in my opinion, the visualzation of more results could increase the quality of the paper materials perception.

Author Response

Dear Reviewer, 

Thank you for your suggestion of my manuscript titled “Reliability and Validity of the Korean Version of the High Performance Work System Scale (HPWS-K)” to International Journal of Environmental Research and Public HealthWe have revised submitted manuscript based on your suggestion. We appreciate the time and effort that you on my manuscript. We have highlighted the changes within the manuscript. 

Point 1. The visualzation of more results could increase the quality of the paper materials perception.

[Re:] Thank you gor your thorough review of our paper, we have revised “table 4” to show more results.

(page 8, lines 253-268)